# Recognition of Consumers’ Characteristics of Purchasing Farm Produce with Safety Certificates and Their Influencing Factors

**DOI:** 10.3390/ijerph15122879

**Published:** 2018-12-15

**Authors:** Jianhua Wang, Ziqiu Gao, Minmin Shen

**Affiliations:** 1School of Business, Jiangnan University, Wuxi 214122, Jiangsu, China; 6180909007@stu.jiangnan.edu.cn (Z.G.); 6180909014@stu.jiangnan.edu.cn (M.S.); 2Jiangsu Research Center of Food Safety, Jiangnan Unversity, Wuxi 214122, Jiangsu, China

**Keywords:** pork with safety certificate, purchase intention, theory of planned behavior (TPB), structural equation model

## Abstract

In order to alleviate the situation that bad money drives out good in the produce market within the context of incomplete information, as well as bridge the gap between demand and product surplus, establishing and improving the safety certification system for farm produce is an urgent need. This paper discusses factors that affect consumers’ purchase of pork with safety certificates in the setting of incomplete information. Data from 844 consumers in Jiangsu and Anhui provinces, along with a structural equation model, are adopted to study consumers’ purchase intention of certified safe pork form. According to our studies, major factors refer to degree of understanding, degree of concern, recognition ability, government publicity, pork’s origin information, consumers’ educational levels, income levels, and consumers’ evaluation of government supervision. Accordingly, suggestions are provided as follows. Above all, enhancing education and training of food safety is conducive to lead consumers’ behaviors in a correct way. Next, news media and social public opinions can play a stronger role in guidance and supervision. Thirdly, an upgraded legal system should be accompanied by better policy implementation. Finally, strengthening the origin certification system and promoting a sense of brand are of significance.

## 1. Introduction

Food is crucial to survival, daily life, and development, so its safety is a great global concern. As a developing country with the largest population, China has a very complex food safety issue, which is closely related to social stability and government authority. As the saying goes, hunger breeds discontentment and food safety is a top priority. China has stepped into a critical transformation stage that greatly satisfies citizens’ demand in food. However, food safety incidents in recent years, such as “melamine”, “scophthalmus maximus”, “Beta-adrenergic agonist”, and “Sudan”, have posed great challenges to consumers’ psychological endurance and confidence in food. Even in developed European countries, consumers still cannot totally trust the existing food supply chain [1]. It is indeed a global issue, which equals a greater eagerness for safe food [2]. From the perspective of economics, unequal information of specific attributes and characteristics of food between consumers and suppliers is the fundamental cause of market failure [3]. To cope with it, third-party certification can serve as a gateway for the provision of more information to consumers, which also changes the information asymmetry to a certain extent [4]. Therefore, the information provided by the third party is vital to consumers. Moreover, the international community is actively exploring new systems and measures to manage and regulate the food industry. As consumers are the ultimate goal of the food production chain and drivers to realize values in the food market [5], their attitudes towards food safety and willingness to consume are greatly influential in future choices of government and food producers. Thus, it is of importance to study the consumption intention of safe food from the perspective of consumers.

In particular, pork consumption in China ranks first in the world, with a value of more than 60% in meat sales in the long-term. As a result, pork is selected as the object of study. Sample data was collected from consumers in Jiangsu and Anhui provinces to establish a corresponding model, so as to analyze how consumers’ attitude toward behavior, subjective norms, and perceived behavioral control affects their purchase intention, as well as to conduct regression analysis on the influence of social characteristics on purchase intention. After combing and analyzing the existing literature, it is found that existing research is still based on the analysis of basic elements, including gender, age, education level, and so on. The biggest contribution of this research is the use of the Kotler behavior selection model to classify consumer characteristics and analyze the main factors affecting consumers’ purchase of safely certified pork. At last, based on the conclusions drawn from the research, recommendations are made to consumers, governments, and food industry entities to improve and create a better food safety environment.

## 2. Literature Review

Previous research has analyzed the purchase of certified safe farm produce, a decision of selection, from various dimensions. Foreign scholars started earlier in this respect. Relevant foreign documents insist on the idea that consumers’ willingness to buy can be summarized as attitude, income, age, education level, social background, subjective norms, and origin information.

In his study of Italian consumers, Magistris T found that attitudes towards organic food determine their final decisions [6]. Furthermore, Chryssohoidis and Krystallis (2005) pointed out that the health benefits and taste of organic foods were taken into consideration [7]. Other scholars believe that there are positive relations between willingness and final purchase. However, willingness varies significantly according to different income levels, ages, education, and regions [8,9], with education as the most influential factor. Besides, Voon J. P. et al. analyzed the influence of emotions and morality on purchase intention in 2001. According to their study, consumers tended to follow influentials’ choices due to Malaysia’s advocacy of collectivism and culture of power [10]. Moreover, other studies have also shown that environmental advocates, advertisements, friends, society, and other elements subordinate to subjective norms are all factors affecting consumers’ willingness to pay [11]. Anna Claret et al. (2012) found that Spanish consumers would select seafish by considering certified origin information, accesses, storage conditions, and prices [12]. Among all the factors, origin information came first. Another study suggested that Tanzanian consumers preferred organic tomatoes to traditional ones, especially those produced domestically [13]. Loyd J.S Baiyegunhi (2018) found that socio-demographic factors such as gender, education, number of children in a household, high income, and race are statistically significant in explaining consumers’ WTP for organic food [14]. Some Chinese scholars have aimed to investigate Chinese consumers’ willingness-to-pay (WTP) for pork, and in Jianhua Wang ‘s study, the Random Parameter Logit model shows that better educational attainment and a higher income were two factors that were associated with a higher WTP [15]. Through the combing of the above documents, it can be found that the analysis of the main factors affecting the purchase of foods for consumer safety certification has been fragmented, and has not been classified and reanalyzed according to a certain logical framework. The advantage of this paper is that the consumer characteristics that affect consumer safety certification pork purchases are divided into four categories before data analysis, and then more logical and dialectical summary.

In contrast, domestic research mainly focuses on individual cases of certain regions rather than a wide range. The reason for this is that subject to objective factors, a relatively small sample size is common in current studies of certified safe produce in China. Generally speaking, existing research is mainly based on surveys targeting a single city, so as to discuss consumers’ views, purchase intention, and influencing factors.

Since Ajzen’s proposal, the complete and mature theory of planned behavior (abbreviated TPB) has been widely applied to various fields of research, such as the pharmaceutical industry, online services, human health research, and agriculture. At the very beginning, this theory was adopted in studying human health. Through this theory, Linda K conducted research on the process of quitting smoking. The results showed that attitude and subjective norms which took intention as media and predicted the progress of the change had important theoretical and practical significance in helping smokers kick off [16]. Furthermore, researchers have applied TPB theory to study how to reduce the incidence of cervical cancer [17]. As this theory is powerful in predicting behaviors, it has already been used to study behavioral intentions. Therefore, this paper is a further study of how to decide on purchase from the perspective of safety certification. Furthermore, Tarkiaimen’s (2005) study also demonstrated that subjective norms had an indirect influence on green consumption intention through attitude [18]. In addition, Vemeira (2008) found that Belgian young people’s consumption intention of sustainable food could be explained by the same three factors as mentioned above [19]. What is more, applications to predict green consumption behaviors by many scholars have confirmed its robustness [20,21,22,23]. In particular, Golnas Rezai et al. (2013) discussed Malaysian consumers’ willingness to pay based on this theory, indicating that their attitude toward behavior, subjective norms, and perceived behavioral control were crucial to the final payment [10]. The results of studies conducted by Kim (2011) suggested that the aforementioned three factors had a positive effect on consumers’ green consumption intention [7].

## 3. Methodology

The TPB theory originated from the multiattribute model [24], which was then developed into the theory of reasoned action and finally into the model we see today [25].

The theory defines attitude toward behavior, subjective norms, and perceived behavioral control, which are the three dimensions that affect behavioral intentions. The three factors have already been confirmed to directly affect final consumption behaviors [26,27]. Generally, scholars will take attitude toward behavior (for example, awareness of health and environment), together with trust in claims of organic food and merits of these food (such as taste, quality, and freshness), into consideration [28,29,30,31,32]. In accordance with McClelland’s “need for achievement” theory, individuals tend to behave in the way that is considered appropriate by their relatives or working groups as the theory involves relations of personal needs and group identification. Similarly, an analysis of 1000 Spanish consumers organic food conducted by Briz T and Ward R W made the conclusion that consumers’ understanding is based on perception and knowledge [2], both of which help to improve consumers’ purchase intention for organic products. Here are three hypotheses therefrom:

**Hypothesis** **H1:**
*Consumers’ attitudes toward certified pork positively affect their willingness to buy.*


**Hypothesis** **H2:**
*Consumers’ subjective norms on pork with safety certification positively affect their purchase intention of such pork.*


**Hypothesis** **H3:**
*Consumers’ perceptual behavior control on certified pork has a positive influence on their purchase intention of such pork.*


The next point is the expectancy-value theory adopted in this study. The basic hypothesis of this theory is that the process of doing something, or participation behavior, depends on the possibility of a behavior-oriented goal and its subjective value [22]. The existing theory in academia can be divided into two schools, namely early Atkinson’s theory and the modern version [31]. In the hypothesis proposed by Eccles, expectations and values were assumed to be influenced by specific beliefs, which also guides model building in this paper. Beliefs describe consumers’ subjective ideas about the various attributes of safe certified agricultural products and affect consumers’ decision-making. Moreover, Brady et al. found that consumers would buy more safe food which was considered as beneficial to physical health [6]. Specifically, the belief factor in this paper is divided into behavioral, normative, and control factors, which also affect behavioral attitudes, subjective norms, and perceived behavioral control. Hence, the following hypotheses are proposed:

**Hypothesis** **H4:**
*Consumers’ behavioral beliefs about certified safe pork are positively correlated to and determine their behavioral attitudes.*


**Hypothesis** **H5:**
*Consumers’ normative beliefs about certified safe pork are positively correlated to and determine their subjective norms.*


**Hypothesis** **H6:**
*Consumers’ control beliefs about certified safe pork are positively correlated to and determine their perceived behavioral control.*


In light of Nayga’s (1996) research, social demographic characteristics of consumers will affect their access to information, their attitudes, and ultimately, their decisions. As a result, this paper also takes the social characteristics of consumers as influencing factors of final behavior. In definition, consumers’ decision-making refers to the process of judging and choosing products, brands, or services ahead of the final purchase. At present, five available models in this aspect are the S-O-R model, Kotler’s behavioral preference model, Nicosia model, Engel model, and Howard-Schells model. It should be noted that Kotler’s behavioral preference model is adopted in this paper as the basis to categorize social characteristics into cultural, social, personal, and psychological types when analyzing the influence of these characteristics on purchase intention. The corresponding hypothesis is as follows:

**Hypothesis** **H7:**
*Consumers’ personal and social characteristics affect their purchase intention of pork with safety certificates, but effects vary according to different characteristics.*


### 3.1. Research Method

The model used in this study is the Structural Equation Model (SEM), also known as the model of latent variables. In other words, both observable and latent variables are commonly seen. Latent variables are especially hard to measure by traditional methods facing the relations among multiple reasons and results in the fields of social science, market, economy, and management. Therefore, the structural equation model was then developed as an important tool for multivariate data analysis. As an alternative for factor analysis, path analysis, multiple regression analysis, and covariance analysis, the model comprises the processes of establishment, estimation, and testing of the causal model. Thus, clear data of a single index can be used to analyze the overall and mutual effects of a single index. Specifically, this model can refer to the measurement model and structural model:

The measurement model, or confirmatory factor analysis model, mainly describes the relations between observable and latent variables by two general equations, namely:x = Λxξ + δ(1)
y = Λyη + ε(2)
where, η is an internal latent variable with the order of n × 1 and y is an internal observable variable with the order of q × 1. Moreover, ξ refers to an external latent variable with the order of m × 1 and x to an external latent variable with the order of P × 1. In addition, Λx is the p × m-ordered matrix, which means the factor loading matrix of y on η. What is more, δ is the p × 1-ordered vector of measurement error and ε is the q × 1-ordered one.

The Structural Equation Model, or casual model of latent variables, is frequently used to discuss correlations of internal and external latent variables. More details are as follows:η = Bη + Γξ + ζ(3)
where, B is the coefficient matrix of internal latent variable η and Γ is the coefficient matrix of external latent variable ξ. The model is also the path coefficient matrix of external and internal latent variables, with ζ as the residual vector.

In summary, the main parameters in this paper are the structural equation coefficient of external and internal latent variables, as well as the measurement model of observable and latent variables.

### 3.2. Data Sources

Data was collected through questionnaires conducted in Jiangsu and Anhui provinces, respectively, from July to September, 2017. Both are located in East China, but the two provinces vary greatly in economic level and lifestyle, so the results could be more objective. According to the principle of hierarchical design, some cities in southern Jiangsu (Suzhou, Wuxi, Changzhou), central Jiangsu (Nantong, Yangzhou, Taizhou), northern Jiangsu (Huaian, Suqian, Xuzhou), southern Anhui (Xuancheng), central Anhui (Hefei), and northern Anhui (Bengbu) were selected as the survey sites. Previous to surveys, professional research experts trained investigators, who conducted random sampling and interviews of 20–30 min at large farmers’ markets, supermarkets, and franchised stores with agricultural products. A total of 844 valid questionnaires were collected from 984 original ones sent out, so the validity rate is 85.77%. Among them, the number of valid questionnaires was 475 from Jiangsu province and 369 from Anhui province.

### 3.3. Analysis of Sample Characteristics

In terms of gender, as shown in Table 1, among 844 interviewees, 473 are female, accounting for 56%, indicating that the females outnumber the males, but little difference is seen between the two groups. From the perspective of age, the largest group accounts for 30.2%, who are mainly under 30 years old. Then interviewees ranging from 40 to 49 years come next, reaching 26.4%. As the young have a more positive attitude towards being surveyed and accepting the safety certification system, they are the majority of interviewees in this survey. In addition, people within the age range of 40 to 49 years old are more easily exposed to surveys owing to their role of major bread-winners and main buyers. As for educational level, among all the interviewees, people with a Bachelor’s degree total 240, ranking first in number. Compared to those with lower degrees, these people are more in touch with and are more concerned about the concept of food safety, so they are more willing to accept the concept and interviews. Besides, 45.9% families are of three members, which is in line with China’s current basic national conditions. The last point is annual household income. People’s living standards are improving with economic growth; 87.4% of interviewees have an annual income of more than 50,000 yuan, which is also closely related to the survey areas we chose.

## 4. Model Estimation and Result

### 4.1. Model Building

The structural equation model adopted in this paper takes consumers’ purchase intention for certified safe pork as an external latent variable, along with attitude to behavior, subjective norms, and perceived behavior control as internal latent variables. In addition, this study also focuses on whether the sociological characteristics of consumption will have an influence on the final purchase intention. Therefore, according to Kotler’s behavioral preference model, this study refers to Philip Kotler’s (1960) analysis on influencing factors of purchase intention and divides consumers’ social characteristics into four major categories, as shown in Table 2.

### 4.2. Exploratory Factor Analysis Along with Tests of Reliability and Validity

SPSS18.0 (IBM, Armonk, NY, USA) and AMOS22.0 (IBM, Armonk, NY, USA) are used to conduct a series of factor analyses on the collected sample data in this paper. To be precise, SPSS18.0 is mainly adopted to conduct exploratory factor analysis and a test of reliability, while AMOS22.0 is employed to test for validity.

#### 4.2.1. Exploratory Factor Analysis

Before factor analysis, SPSS18.0 is used to test its suitability, including Kaiser-Meyer-Olink Measure of Sampling Adequacy and Bartlett Test of Sphericity. The corresponding factor loading matrix gained is shown in the following table. The results show that the KMO value of all data is 0.740 and the scale’s KMO value is basically greater than 0.6. In addition, Bartlett’s test is significant and the Sig value of 0.000 < 0.05. Moreover, the standard factor loading coefficient of each tested variable is bigger than 0.6. This indicates that factor analysis is applicable to the sample data in this paper.

#### 4.2.2. Test of Reliability and Validity

A reliability test is used to check questionnaires’ degree to reflect real situations. Data analysis of scale is valid only when reliability is confirmed [27]. In this paper, the major method to test reliability is to measure the results’ consistency through Cronbach’s α coefficient. The rule is that the higher the α coefficient is, the better the consistency is. In accordance with past experience, a value above 0.9 equals a very high reliability, a value between 0.8 and 0.9 equals a relatively high reliability, a value between 0.7 and 0.8 equals a relatively high reliability, and a value between 0.6 and 0.7 equals an acceptable reliability. In this paper, the Cronbach’s α coefficient is 0.876 for the scale and about 0.7 for each item, which means that the variables are highly consistent. The results of the validity test are shown in Table 3.

### 4.3. Test Results and Analysis

#### 4.3.1. Analysis of Fitting Test and Path Coefficient

This paper adopts AMOS 22.0 to test the latent variables in the model, including attitude to behavior, subjective norm, perceptual behavior control, purchase intention, and their corresponding observable variables. The fitting values obtained are shown in Table 4. The results are in an ideal state as the data fits the model well.

Figure 1 is a variable coefficient path diagram obtained after regression analysis based on AMOS 22.0. The meaning of the specific image will be elaborated in Section 4.3.2.

#### 4.3.2. Path Analysis of Structural Equation Model

According to the standardization model coefficients, the following structural model Equation (1) and measurement model Equation (2) are deducted. Please refer to Figure 1 for more details:Purchase Intention = 0.408 × perceived behavior control + 0.268 × subjective norms + 0.531 × attitude to behavior + 0.367 × attitude to behavior × subjective norms + 0.4 × subjective norms × perceived behavior control + 0.22 × attitude to behavior × perceived behavior control + e(1)
(2)$$\left[\begin{array}{c} WISE\\ SUPP\\ MEAN\\ EXPE\\ RECO\\ CHAN\\ COST\\ MARK\\ CONS\\ GOVE\\ MEDI\\ ACIN \end{array}\right]=\left[\begin{array}{ccc} 0.729 & 0 & 0\\ 0.876 & 0 & 0\\ 0.714 & 0 & 0\\ 0 & 0.564 & 0\\ 0 & 0.537 & 0\\ 0 & 0.699 & 0\\ 0 & 0.653 & 0\\ 0 & 0.556 & 0\\ 0 & 0 & 0.689\\ 0 & 0 & 0.736\\ 0 & 0 & 0.513\\ 0 & 0 & 0.48 \end{array}\right]\times \left[\begin{array}{c} \mathrm{attitude}\;\mathrm{of}\;\mathrm{behavior}\\ \mathrm{perceived}\;\mathrm{behavioral}\;\mathrm{control}\\ \mathrm{subjective}\;\mathrm{norm} \end{array}\right]+\left[\begin{array}{c} e1\\ e2\\ e3\\ e4\\ e5\\ e6\\ e7\\ e8\\ e9\\ e10\\ e11\\ e12 \end{array}\right]$$

Equation (1) reflects the correlations of latent variables. It indicates that consumers’ attitude to behavior, subjective norms, and perceived behavior control positively affects their willingness to buy. The more positive the consumers’ attitudes are, the more they will be influenced by the external factors. When consumers’ control of personal behaviors reaches a certain extent, their purchase intention is stronger. Judging by the final results of the path coefficient, the three latent factors above have a positive influence on each other. Therefore, the conclusion can be drawn that hypotheses H1, H2, and H3 are true.

In contrast, Equation (2) reveals the relations of observable and latent variables as follows:(1)In terms of attitude to behavior, consumers support the purchase of certified safe pork. The path coefficient of support purchase is the largest and the path value of considering purchase as wise ranks second.(2)In consumers’ minds, easy characteristics variables to recognize certified pork are the most significant among variables of perceived behavior control as the corresponding path coefficient is 0.736. It also indicates whether pork with certificates has a great influence on their purchase decisions. The next influential factor is that adequate experience can ensure the safety of purchased pork, with a path coefficient of 0.689. Two other observable variables also have similar positive influence on latent variables in the aspect of perceived behavior control, namely the cost to purchase certified safe pork has not been raised and convenient channels to buy are available.(3)Moreover, government promotion and support as a measurable variable has the greatest influence on consumers’ subjective norms, showing a path coefficient of 0.699. This means that the government plays a crucial role in consumers’ life. As a result, stronger supervision and management by the government can improve information asymmetry, so as to increase consumers’ confidence in certified safe pork and further promote their purchase. Another major factor is the media, with a path coefficient of 0.653. The combination of we-media and mainstream media boosts the prosperity of this industry and thus has a positive influence on consumers. Besides, family relatives, friends, colleagues, bosses, experts, and academic institutions have demonstrated a positive and balanced effect on consumers, while other consumers are the least influential to consumers’ subjective norms.

### 4.4. Regression Analysis

In view of the regression analysis results in Table 5, each independent variable has a great influence on dependent variables, which also verifies the rationality of hypotheses H4, H5, and H6. From the perspective of the influence of behavioral beliefs on behavioral attitudes, consumers believe that the origin information of certified safe pork demonstrates the strongest influence on behavioral attitudes. In other words, it also reflects that the origin, the non-sensory information attribute used to represent quality and safety, is vital to consumers’ choices [33]. Since the “melamine incident”, the Chinese prefer imported milk powder as they have doubts about domestic products. In that way, they will pay more attention to the place of origin. Similarly, in the pork market, as the Chinese consume the largest amount of pork around the world, it is impossible to rely on imported pork. Therefore, origin certification is urgent to ensure pork safety. In the setting of normative beliefs’ influence on subjective norms, advice from relatives and friends and suggestions from bosses and colleagues are of similar importance, possibly owing to the similar time of contact and expectations. However, when considering the influence of control beliefs on perceived behavior control, government supervision is a decisive factor that affects final purchase, indicating the government’s active role in leading consumption.

Generally speaking, individual factors, including age, gender, and marital status, have no significant influence on attitudes to behavior, subjective norms, and perceived behavioral control in this study. It also indicates that the gap in these three aspects in narrowing down. According to Table 6, in terms of cultural factors, education level, income level, and family population are significantly influential to subjective attitudes, but have less influence on subjective norms. Only the education level is highly decisive to perceived behavior control; that is, a higher education level means a stronger ability of self-perception and control. Considering social factors, local safety conditions and degree of understanding are more dominant in deciding attitudes to behavior and perceived behavioral control. But for subjective norms, the major factor is simply degree of understanding. That is to say, the influence of external factors is based on consumers’ knowledge of certified safe pork. Psychologically, degree of concern over pork safety and degree of satisfaction with government supervision have a positive influence on the above three aspects, but consumers’ experience of safety incidents shows a negative and obscure influence. This analysis can partially justify hypothesis H7. In other words, social characteristics have different effects on purchase intention. All in all, social characteristics’ significant influence mainly focuses on cultural, social, and psychological sides. Therefore, enterprises should take the rule into consideration when making marketing strategies.

## 5. Conclusions

Chinese consumers are paying increasing attention to food safety issues, and these concerns can be attributed to many aspects of the impact. Based on the theory of planned behavior and Kotler’s behavioral preference model, this paper adopts a structural equation model to analyze factors that affect consumers’ purchase of certified safe pork. The results show that attitude to behavior, subjective norms, and perceived behavior control positively influence purchase intention, with the first as the greatest influencing dimension. From the internal mechanism, consumers supporting the attitude of purchasing safety certification and consumers with a sufficient experience of buying safety certification pork are the two internal factors that affect consumer safety certification pork purchase intention. In terms of the external mechanism, the influence of the government and the media on consumers is the most significant. In terms of subjective norms, promotion by the government and media has the most significant influence. In addition, cognition ability and experience dominantly affect perceived behavior control. Moreover, origin information is crucial to final decisions when discussing beliefs’ influence. That is to say, more trust in pork from one origin is equal to stronger purchase intention. Government supervision demonstrates its importance as it is vital in the aspect of control belief. According to the research gap of the existing research, the existing research only points out the social characteristics of consumers that affect the consumer safety certification of pork purchase behavior, and does not classify and summarize these influencing factors. In the light of comprehensive regression analysis of social characteristics’ effect on the above three dimensions, cultural, social, and psychological factors matter a lot. Specifically, factors include consumers’ educational level, their income, the quality and safety of pork in the place where consumers live, consumers’ understanding of pork safety certification, their concern about pork safety, and their satisfaction with government supervision.

Hence, corresponding suggestions can be put forward as follows. To start with, enhancing education on and training of food safety is conducive to lead consumers’ behaviors in a correct way. It is mainly demonstrated in improving consumers’ legal awareness, understanding of food safety, and degree of concern on this issue, so as to lead consumers to safeguard personal rights. The next point is that news media and social public opinions can play a stronger role in guidance and supervision. That is to say, the news media should convey more effective information that contributes to better judgement and decisions. In addition, they should bear the supervision responsibility to disclose illegal business practices in order to regulate production processes of the agricultural product industry. Thirdly, an upgraded legal system should be accompanied by better policy implementation, specifically, stricter market access and more severe punishments of violations. Finally, strengthening the origin certification system is of significance. As consumers have varied preferences of origin certification, it has become an important measurement for market decisions and positioning. Therefore, one priority of current major players in the food industry is to improve origin certification and enhance the sense of brand, so as to reduce consumers’ selection costs and increase efficiency.

## Figures and Tables

**Figure 1 ijerph-15-02879-f001:**
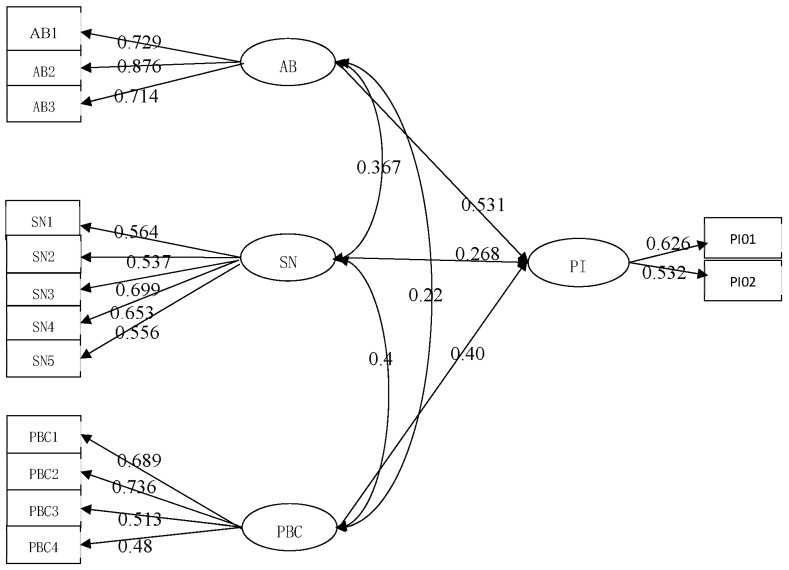
Structural equation model variable regression path coefficient graph.

**Table 1 ijerph-15-02879-t001:** Descriptive statistics of consumer social demographic characteristics.

Social Demographic Characteristics	Frequency	Percentage	Total Percentage	Social Demographic Characteristics	Frequency	Percentage	Total Percentage
sex				Family member			
Female	473	56	56	1	6	0.7	0.7
Male	371	44	100	2	70	8.3	9
Age				3	387	45.9	54.9
30 or below	255	30.2	30.2	4	173	20.5	75.4
30-39	151	17.9	48.1	More than 5	208	24.6	100
40-49	223	26.4	74.5	Family annual income			
50-59	134	15.9	90.4	50,000 or below	106	12.6	12.6
60 or above	81	9.6	100	50,000–80,000	183	21.7	34.2
marital status				80,000–100,000	230	27.3	61.5
unmarried	222	26.3	26.3	100,000 or above	325	38.5	100
married	622	73.7	100	Is there a child under 18?			
degree				No	431	51.1	51.1
Junior high school or below	237	28.1	28.1	Yes	413	48.9	100
High school (including secondary occupation)	208	24.6	52.7	Is it the main purchaser of household daily food?			
College	113	13.4	66.1	No	390	46.2	46.2
Bachelor	240	28.4	94.5	yes	454	53.8	100
Graduate student or above	46	5.5	100				

**Table 2 ijerph-15-02879-t002:** Design of consumer social feature items based on Kotler’s behavior selection model.

Classify	Number	Items
	AGE	age
individual factors	GEND	sex
	EDU	Education degree
cultural factors	INCOME	Family annual income
	NUMBER	Family population
Social factors	STATUS	Current quality and safety status of pork
	KNOW	Safety certification pork understanding
	CONCERN	Concern about pork safety issues
psychological factors	ENCOUNTER	Whether to encounter the problem of pork quality safety
	DEGREE	Satisfaction with the effectiveness of government regulation

**Table 3 ijerph-15-02879-t003:** Test of validity.

latent Variable (Code)	Observable Variables (Code)	Cronbach’α	Factor Loading	Bartlett Test of Sphericity	KMO Sample Measure	C.R	AVE
attitude of behavior (AB)	I think it’s wise to buy safe and certified pork (AB1)	0.753	0.694	678.627 (P = 0.000)	0.66	0.821	0.606
	I support the purchase of safe and certified pork (AB2)		0.825				
	I believe that implementing pork quality safety certification can increase consumer confidence in food safety (AB3)		0.809				
subjective norm (SN)	Family, relatives and friends have a great influence on my purchase of safe certified pork. (SN1)	0.738	0.682	803.192 (P = 0.000)	0.794	0.834	0.503
	Colleagues have a great influence on my purchase of safe certified pork. (SN2)		0.575				
	The government’s publicity call has a big impact on my purchase of safely certified pork. (SN3)		0.641				
	The media information has a great impact on my purchase of safe certified pork. (SN4)		0.774				
	The opinions of experts and academic institutions are very big for me to buy safe certified pork. (SN5)		0.563				
perceived behavioral control (PBC)	I have enough experience to ensure the safety of the pork I purchased (PBC1)	0.693	0.782	594.522 (P = 0.000)	0.692	0.808	0.514
	I think it is not difficult to identify the characteristics of safely certified pork at the time of purchase. (PBC2)		0.78				
	For me, it’s convenient to buy safe certified pork. (PBC3)		0.525				
	For me, the cost of purchasing safely certified pork has not increased significantly. (PBC4)		0.568				
purchase intention (PI)	Do you have the idea of purchasing a safe certified pork? (PI1)	0.601	0.606	199.095 (P = 0.000)	0.5	0.752	0.504
	Have you purchased safety certified pork in your daily life? (PI2)		0.749				

**Table 4 ijerph-15-02879-t004:** Overall fitness evaluation standard and fitting evaluation result of structural equation model.

Index Category	Index Name	Evaluation Standard	ACTUAL FIT	Compared with Evaluation Criteria	Result
Absolute fit index	χ2/df	<3	2.832	<3	Ideal
	GFI	>0.90	0.968	>0.90	Ideal
	RMR	<0.05	0.037	<0.05	Ideal
	RMSEA	<0.05	0.044	<0.05	Ideal
Incremental fitness index	NFI	>0.90	0.934	>0.90	Ideal
	IFI	>0.90	0.958	>0.90	Ideal
	TLI	>0.90	0.945	>0.90	Ideal
	CFI	>0.90	0.957	>0.90	Ideal

GFI (Goodness of Fit Index); RMR (Root Mean square Residual); RMSEA (Root Mean square Error of Approximation); NFI (Nprmed Fit Index); IFI (Incremental Fit Index); TLI (Tucker-Lewis Index); CFI (Comparative Fit Index).

**Table 5 ijerph-15-02879-t005:** Regression analysis of beliefs on the three dimensions of planned behavior theory.

Dependent Variable	Independent Variable	β	Standard β	T Value	R2	F Value
attitude of behavior	(constant)	−2.308 ***		−16.089	0.244	90.33
	BB1	0.146 ***	0.038	3.835		
	BB2	0.089 ***	0.039	2.293		
	BB3	0.396 ***	0.039	10.247		
subjective norm	(constant)	−1.497 ***		−14.879	0.244	135.598
	NB1	0.218 ***	0.257	7.421		
	NB2	0.247 ***	0.312	9.014		
perceived behavioral control	(constant)	−0.819 ***		−5.479	0.037	16.263
	CB1	0.094 ***	0.108	3.086		
	CB2	0.141 ***	0.134	3.832		

Note: *** indicates significant at >0.001

**Table 6 ijerph-15-02879-t006:** Regression analysis of consumer characteristics on three dimensions of planned behavior theory and consumer intention.

Dependent Variable	Classify	Independent Variable	β	Standard β	T Value	R2	Adjust R2	F Value
attitude of behavior		(constant)	−2.224 ***		−8.769	0.195	0.184	18.311
	Individual factors	Gender	0.067	0.033	1.067			
		Age	0.034	0.045	0.961			
		Marital	−0.05	−0.022	−0.501			
	cultural factors	Education	0.082 **	0.108	2.586			
		Family member	−0.085 **	−0.082	−2.589			
		income	0.051	0.053	1.567			
	Social Factors	status	0.221 ***	0.197	4.892			
		degree	0.138 ***	0.133	3.32			
	Psychological factors	know	0.181 ***	0.176	5.331			
		concern	0.121 ***	0.124	3.729			
		encounter	−0.016	−0.005	−0.154			
subjective norm		(constant)	−1.536 ***		−5.776	0.115	0.104	9.857
	Individual factors	Gender	−0.035	−0.018	−0.534			
		Age	−0.058	−0.077	−1.578			
		Marital	0.063	0.028	0.599			
	cultural factors	Education	−0.02	−0.026	−0.598			
		Family member	0.012	0.012	0.363			
		income	0.004	0.004	0.103			
	social factors	status	0.068	0.061	1.434			
		degree	0.220 ***	0.213	5.062			
	Psychological factors	know	0.131 ***	0.128	3.684			
		concern	0.108 **	0.11	3.167			
		encounter	−0.013	−0.004	−0.116			
perceived behavioral control		(constant)	−2.996 ***		−12.074	0.229	0.219	22.495
	Individual factors	Gender	0.084	0.042	1.361			
		Age	0.032	0.043	0.942			
		Marital	0.125	0.055	1.288			
	cultural factors	Education	0.072 *	0.094	2.311			
		Familymember	0.06	0.058	1.871			
		income	0.092 **	0.096	2.906			
	social factors	status	0.189 ***	0.169	4.271			
		degree	0.143 ***	0.139	3.522			
	Psychological factors	know	0.260 ***	0.253	7.833			
		concern	0.101 **	0.103	3.173			
		encounter	−0.073	−0.022	−0.707			
purchase intention		(constant)	−2.425 ***		−9.753	0.226	0.216	22.095
	Individual factors	Gender	−0.114	−0.056	−1.836			
		Age	−0.032	−0.042	−0.916			
		Marital	0.041	0.018	0.416			
	cultural factors	Education	0.083 **	0.108	2.655			
		Familymember	−0.001	−0.001	−0.024			
		income	0.126 ***	0.132	3.979			
	social factors	status	0.146 ***	0.13	3.299			
		degree	0.053	0.051	1.305			
	Psychological factors	know	0.266 ***	0.259	7.992			
		concern	0.163 ***	0.167	5.11			
		encounter	0.049	0.015	0.475			

Note: *** indicates significant at >0.001, ** indicates significant at >0.01, and * indicates significant at >0.05.

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
