# Peer review of "Recognition of Consumers’ Characteristics of Purchasing Farm Produce with Safety Certificates and Their Influencing Factors"

_ijerph, 2018, doi:10.3390/ijerph15122879_

Round 1

Reviewer 1 Report

(1) In the "Introduction" section, I would suggest that the authors could clearly point out the research gaps and research contributions of the present study.

(2) I would suggest that the authors could compare some of the findings with the existing studies, since the present study is an extension of the present studies.

(3) when the authors define the symbols in the equations, the sentence should begin with "where".

Author Response

Response:

1From line 52 to line 56we added the research gaps and research contributions of the present study.

2From line 351 to line 358, we compare some of the findings with the existing studies and present the conclusions of the study based on the gaps in the existing research.

3we added “where” before the symbols in the equations, See line 174 and line 182 for details.

Reviewer 2 Report

Thank You very much for the possibility to become familiar with an interesting article. It is very well-written and has a research character. The Authors should be appreciated for the research reliability and proper selection of data analysis methods. However, because the best is the enemy of the good, I suggest some improvements in that text.

1. The Authors presented a too poor overview of the literature. It should be expanded by the works of other researchers, which show the whole spectrum of the considered problem. Especially, the paper lacks references to the latest articles from years 2017-2018.

2. I suggest the wider description of the way of selection of the examined sample. In my opinion, it should be emphasised if the selection was intentional or random. Even with the most accurate selection of the data analysis methods, the results obtained from nonrandom samples should be treated with caution.

3. It should be explained why the internet survey method was used. Why did the Authors consider it to be suitable, despite its limits and disadvantages?

4. The explanation of the size of the sample is very laconic. The sample is quite small, especially for such a big country like China. This is another reason why the results should be treated with caution.

5. The diagram which is placed just after the Table 5 should be presented as Figure 1. The Authors have to care about the readability of numerical values presented in the diagram.

6. Why did the Authors place Table 6 (line 261), despite the fact that they do not refer to it in the text?

7. Two different tables have number 6: line 261 and 298.

8. References contain numerous mistakes. The Authors should pay more attention to them.

9. The editing and formatting in the whole paper should be corrected in order to make it clearer.

Author Response

Response:

1A review of the latest literature from year 2017 to 2018 has been added between lines 77 and 87. And on the basis of the addition, logical dialectical thinking is carried out.

2. From line 194 to line196, this survey arranges professional research experts to conduct pre-investigation training for investigators to ensure the reliability and accuracy of the survey data. The survey selects large farmers' markets in various places through professionally trained investigators. , large supermarkets and franchised stores of agricultural products, through random sampling, interviews, etc.,

3. The article does not use the Internet survey method. It is based on the design of the questionnaire. The investigators go to the actual large-scale farmer's market, large supermarkets, and agricultural product franchise stores to conduct surveys through random sampling and interviews.

4. Line 88 mentioned that domestic researches mainly focus on individual cases of certain regions rather than a wide range.Due to the large geographical area of China, it is very difficult to conduct a national survey. Therefore, the best way is to select several of them to conduct comparative surveys. Therefore, this survey selected two countries with large differences in economic levels. In order to ensure the validity of the comparison, the sample size of the two provinces is guaranteed to be the same in selecting the survey sample. Because of the funding and time constraints, 844 valid samples were used.

5.An explanation was added on line 260.

6.The table has been deleted.

7. The number in the second table 6 is incorrectly marked.

8. Reference errors have been corrected one by one

9. We have formatted the manuscript to journal style including a well demarcated (1) introduction, (2) literature review , (3) methodology (4) Model Estimation and Result (5) Conclusion and Policy Implications

Reviewer 3 Report

The manuscript present important finding on the relationship between consumers attitude, subjective norms and behavior towards the purchase of food with safety certificates. However, authors failed to format the manuscript to journal style including a well demarcated (1) introduction, (2) material and methods (3) result (4) discussion (5) conclusion. Such demarcations make for clarity and ease of reading.

Other weakness of the manuscript would be

1.       Widespread grammatical errors

2.       Widespread spacing problems

3.       Typographic errors are also common

4.       The introductory part is too long and must be trimmed

5.       The result part contains a lot of repetitions eg statistical analysis

6.       Tables often contains too many details and some showing repeat kind of analysis eg R2 and adjusted R2.

7.       Table 2 should be deleted as these are merely coding that aided in the analysis. Also consider reducing the number of tables in the manuscript

8.       It is possible to have a unified hypothesis

The manuscript should be sent for English editing and significantly shorted and the writing precise.

Specific comments are attached in-text.

Author Response

Response:

We have formatted the manuscript to journal style including a well demarcated (1) introduction, (2) literature review , (3) methodology (4) Model Estimation and Result (5) Conclusion and Policy Implications

1-3: The problem of the text translation part and Typographic errors have been improved.

4. The introduction section was deleted from line 47 to line 58

5.The repetitive part of the conclusion has been deleted and the language has been reorganized. See lines 342 to 347 for details.

6. Adjusted R2 of table 6 have been deleted.

7.Table2 have been deleted.

8. Thank you for your suggestion, You made a good idea, but based on our current review of the literature, we have not found a good unified hypothesis in the literature, but your suggestion is very pertinent, we will achieve this goal in the next study.

Round 2

Reviewer 3 Report

Authors have addressed most of my concerns